

**Characterization and application of artificial light sources for nighttime**
**aerosol optical depth retrievals using the VIIRS Day/Night Band**

Jianglong Zhang[1], Shawn L. Jaker[1], Jeffrey S. Reid[2], Steven D. Miller[3], Jeremy Solbrig[3], and
Travis D. Toth[4]
[1]Department of Atmospheric Sciences, University of North Dakota, Grand Forks, ND, USA
[2]Marine Meteorology Division, Naval Research Laboratory, Monterey, CA, USA
[3]Cooperative Institute for Research in the Atmosphere, Colorado State University, Fort Collins,
CO, USA
[4]NASA Langley Research Center, Hampton, VA, USA

Corresponding Author: jzhang@atmos.und.edu



**Abstract**



Using nighttime observations from Visible/Infrared Imager/Radiometer Suite (VIIRS) Day/Night
band (DNB), the characteristics of artificial light sources are evaluated as functions of observation
conditions and incremental improvements are documented on nighttime aerosol retrievals using
VIIRS DNB data on a regional scale. We find that the standard deviation of instantaneous radiance
for a given artificial light source is strongly dependent upon the satellite viewing angle, but is
weakly dependent on lunar fraction and lunar angle. Retrieval of nighttime aerosol optical
thickness (AOT) based on the novel use of these artificial light sources is demonstrated for three
selected regions (United States, Middle East, and India) during 2015. Reasonable agreements are
found between nighttime AOTs from VIIRS DNB and temporally adjacent daytime AOTs from
AErosol RObotic NETwork (AERONET) as well as from coincident nighttime AOT retrievals
from the Cloud-Aerosol Lidar with Orthogonal Polarization (CALIOP), indicating the potential of
this method to begin filling critical gaps in diurnal AOT information at both regional and global
scales. Issues related to cloud, snow, and ice contamination during the winter season, as well as
data loss due to the misclassification of thick aerosol plumes as clouds, must be addressed to make
the algorithm operationally robust.



# 1    Introduction


The Visible/Infrared Imager/Radiometer Suite (VIIRS), on board the Suomi National Polar-
orbiting Partnership (NPP) satellite, features 22 narrow-band channels in the visible and infrared
spectrum. Included on VIIRS is the Day/Night band (DNB), designed to detect both reflected
solar energy at daytime and low light visible/near-infrared signals at nighttime (e.g., Lee et al.,
2006; Miller et al., 2013; Elvidge et al., 2017). Compared to the Operational Line Scan (OLS)
sensor on the legacy Defense Meteorological Satellite Program (DMSP) constellation, the VIIRS
DNB has improved response to nighttime visible signals, owing to its higher spatial resolution,
radiometric resolution, and sensitivity (e.g., Miller et al., 2013; Elvidge et al., 2017). The DNB,
unlike the OLS, is calibrated which enables quantitative characterization of nighttime
environmental parameters via a variety of natural and artificial light signals, including reflected
moon light in cloudy and cloud free regions, natural and anthropogenic emissions from forest fires,
volcanic eruptions, gas flares from oil fields, and artificial light sources from cities (e.g., Miller et
al., 2013; Elvidge et al., 2017).
Using nighttime observations from VIIRS/OLS over artificial light sources such as cities,
several studies have attempted to derive nighttime aerosol optical properties. For example, Zhang
et al. (2008) propose the concept of estimating nighttime aerosol optical thickness (AOT) by
examining changes in DMSP/OLS radiances over artificial light sources between aerosol free and
high aerosol loading (and cloud-free) nights. However, the OLS visible channel does not have on-
board calibration, which limits the use of OLS data for quantitative studying of nighttime aerosol
properties. VIIRS's improved spatial and spectral resolutions and on-board calibration make
accurate quantification of nighttime aerosol properties feasible.



Using VIIRS radiances over selected artificial light sources, Johnson et al. (2013) develops a
retrieval of nighttime AOT for selected cities. However, radiances from artificial light-free regions
are needed for this retrieval process. McHardy et al. (2015) proposes an improved method, based
on the method proposed by Johnson et al. (2013) which uses changes in spatial variations within
a given artificial light source for retrieving nighttime AOT. The advantage of McHardy et al.
(2015) is that only observations over the artificial light sources themselves are needed, eliminating
the need for artificial light-free regions and implicit spatial invariance assumptions of Johnson et
al. (2013).
As proof-of-concept studies, only a few selected artificial light sources have been considered
in those pioneering nighttime aerosol retrieval studies that utilize VIIRS observations. As
suggested from McHardy et al. (2015), careful studies of the characteristics of artificial light
sources are needed to apply the method over a broader domain. Thus, in this study, using VIIRS
data from 2015 over the US, Middle East, and India, we focus on answering the following
questions:
(1) How do radiance fields from artificial light sources vary as functions of observing

conditions?

(2) Are nighttime AOT retrievals using VIIRS DNB feasible on a regional basis? In

particular, for our selected regions, can reasonable agreement be achieved between

nighttime VIIRS DNB derived AOT, aerosol retrievals from Cloud-Aerosol Lidar with

Orthogonal Polarization (CALIOP), and approximated nighttime AOT values from

daytime AErosol RObotic NETwork (AERONET)?

(3) What are the limitations in the current approach that can be improved in future attempts?



In the current study, we do not aim to finalize the nighttime retrieval methods, but rather
explore existing issues, report incremental advancements, and propose revised methods for future
studies. This paper is organized as follows: Section 2 introduces the datasets used in this study as
well as data processing and aerosol retrieval methods. Section 3 discusses artificial light source
patterns as functions of viewing and lunar geometries and lunar fraction, as well as other
observation-related parameters. Results of regional-based retrievals are also included in Section
3. Section 4 closes the paper with discussion and conclusions.

## 118    2    Datasets and Methods

### 119    2.1 Datasets

Flying in a sun-synchronous polar orbit, VIIRS has a local nighttime overpass time of ~1:30
am. The spatial resolution of a VIIRS DNB pixel is ~750 m across the full swath width of ~3000
km. VIIRS DNB observes at a wavelength range of 0.5 - 0.9 µm, with a peak wavelength of ~0.7
µm (e.g., Miller et al., 2013). VIIRS differs from its ancestor, OLS, by providing on-board
calibration for tracking signal degradation as well as changes in modulated spectral response
function through the use of a solar diffuser (e.g., Chen et al., 2017). Early versions of VIIRS DNB
data suffer from stray light contamination (e.g., Johnson et al., 2013). These issues have since
been corrected for in the later version of the VIIRS DNB data (Mills et al., 2013).
In this study, three processed and terrain-corrected VIIRS datasets were used. The
VIIRS/DNB Sensor Data Record (SVDNB) includes calibrated VIIRS DNB radiance data for the
study as well as Quality Assurance (QA) flags for each pixel. The VIIRS Cloud Cover/Layers
Environmental Data Record (VCCLO) dataset was used for cloud clearing, and the VIIRS/DNB
Sensor Data Record Ellipsoid Geolocation (GDNBO) dataset was used for obtaining geolocation



for the VIIRS DNB radiance data.  The GDNBO dataset also includes other ancillary parameters
including solar, lunar, and satellite zenith/azimuth angles, as well as lunar phase that were used as
diagnostic information in support of this study.

To evaluate the VIIRS retrieved AOTs, cloud-cleared and quality-assured Level 2, Version 3

AErosol RObotic NETwork (AERONET) data were enlisted as the "ground truth."  Reported in
AERONET data are AOTs at a typical wavelength range of 0.34 to 1.64 μm (Holben et al., 1998).
We point out that AERONET AOTs are derived through measuring the attenuation of solar energy
at defined wavelengths, and thus are only available during daytime.  Therefore, averaged AOTs
(0.675 μm) for the day before and after the VIIRS observations were used in evaluating the
performance of VIIRS retrievals at night.  A pair of VIIRS and AERONET retrievals are
considered collocated if the temporal difference is within ± 24 hours and the spatial difference is
within 0.4° Latitude/Longitude.  All collocated AERONET data for one VIIRS data point were
averaged to represent the AERONET-retrieved AOT value of the desired VIIRS retrieval.

Nighttime aerosol retrievals are also available from CALIOP aerosol products at both regional

and global scales and for both day and nighttime (Winker et al., 2007).  Thus, we also inter-
compared VIIRS nighttime AOTs retrieved from this study with CALIOP column integrated
AOTs.  The Version 4.10, Level 2 CALIOP aerosol profile products (L2_05kmAPro) were used
in this study.  Upon quality assurance steps, as mentioned in Toth et al. (2018), column integrated
CALIOP AOTs were derived at the 0.532 and 1.064 μm channels and then interpolated to the 0.70
μm channel (central wavelength of the DNB) for this study.  The VIIRS and CALIOP data pair is
considered to be collocated if the spatial difference was within 0.4° Latitude/Longitude and the
temporal difference was within ± 1 hour.  Note that one VIIRS retrieval may be associated with



multiple CALIOP AOT retrievals, and thus collocated CALIOP aerosol retrievals were averaged
to a single value for this comparison.

An open-source global city database from MaxMind (http://www.maxmind.com/) was used in

this study for cross checking with the detected artificial light sources for this study. The city
database includes the name and geolocation of the cities as well as other ancillary information.
Based on these data, a total of 999 cities from the Middle East region (11-42°N, 28-60°E) and
2995 cities from the India region (8-35°N, 68-97°E) were used in this study. These cities, as well
as their geolocations, are shown in Figs. 1b and 1c for the Middle East and India regions,
respectively, and are documented and attached as appendices to this paper.

One focus of this study is to understand the variations of artificial light sources as a function

of observing conditions. To achieve this goal, we have arbitrarily selected 200 cities across the
US. Since aerosol loadings are relatively low in the US compared to regions such as the Middle
East and India, this selection gives insight into the characteristics of artificial light sources. Also,
we require the selected cities to be isolated – that is, not in the immediate vicinity of another city
or major light source, so as to avoid light dome contamination. The majority of selected cities
have populations within the range of 25,000 and 100,000 with a few higher-population exceptions
such as Memphis, New Orleans, and Charleston. The geolocations of the 200 cities are shown in
Fig. 1a, and as mentioned above, the full list of the cities are also included as an attachment.

**2.2 Retrieval methods**

The theoretical basis for retrieving nighttime aerosol optical depth using stable artificial lights

is based upon previous studies (Zhang et al., 2008; Johnson et al., 2013; McHardy et al., 2015). In



the current approach, the VIIRS-observed radiance over a cloud free artificial light source can be
expressed as:
$$I_{sat} = I_s e^{-\tau/\mu} + I_s \mathrm{T}(\mu) + I_p \tag{1}$$

Where $I_{sat}$ is the satellite received radiance, represented as the sum of contributions from three

principal components: upwelling surface light emission through direct ($I_s e^{-\tau/\mu}$) and diffuse ($I_s \mathrm{T}(\mu)$)
transmittance, and the path radiance source term ($I_p$). Here, $\tau$ is the total column optical depth
from aerosol and Rayleigh components, $\mu$ is the cosine of the viewing zenith angle, and $\mathrm{T}(\mu)$ is
the diffuse-sky transmittance. $I_s$ is the cloud free sky surface upward radiance, which can be
further rewritten as:
$$\pi I_s = r_s (\mu_0 F_0 e^{-\tau/\mu_0} + \mu_0 F_0 \mathrm{T}(\mu_0) + \pi I_s \bar{r}) + \pi I_a \tag{2}$$

Where $r_s$, $\mu_0$, $F_0$ are (respectively) the surface reflectance, cosine of the lunar zenith angle, and

the top-of-atmosphere downward lunar irradiance convolved with the VIIRS DNB response
function. $\mathrm{T}(\mu_0)$ is the diffuse transmittance term, $\bar{r}$ is the reflectance from the aerosol layer, and
$I_a$ is the emission from the artificial light source. The three terms inside the parentheses of Eq. 2
comprise the surface downward irradiance terms, where $\mu_0 F_0 e^{-\tau/\mu_0}$ is the downward irradiance from
moonlight through direct attenuation (or $F_{directdown}$) and $\mu_0 F_0 \mathrm{T}(\mu_0)$ is the downward irradiance from
moonlight through diffuse transmittance (or $F_{diffusedown}$). The $\pi I_s \bar{r}$ term represents the surface
emission (irradiance) that is reflected back downward to the surface by the aerosol layer that has
a layer mean reflectivity of $\bar{r}$. Eq. 2 shows that the surface emission term includes emission from
the artificial light source, as well as from reflected downward fluxes. Solving $I_s$ from Eq. 2,
inserting that result into Eq. 1, and rearranging, yields:
$$I_{sat} = \frac{r_s(F_{directdown} + F_{diffusedown}) + \pi I_a}{\pi(1 - r_s \bar{r})} [e^{-\tau/\mu} + \mathrm{T}(\mu)] + I_p \tag{3}$$







We expect the artificial light source emission term, $I_a$, to vary spatially within a heterogeneous
light source such as a larger city. Within that city, we can assume that the $F_{directdown}$, $F_{diffusedown}$,
and $I_p$ terms have negligible spatial variations. This assumption follows McHardy et al. (2015),
who also assume the surface diffuse emission term ($I_s\ T(\mu)$) is spatially invariant. However, as
indicated in Eq. 3, the surface diffuse emission term includes the $I_s$, which contains the $I_a$ term.
Thus, we retain the surface diffuse emission term in this study.
By taking the spatial derivative of Eq. 3 (using the delta operator $\Delta$) and by eliminating terms
that have small variation within a city, we can derive:
$$\Delta I_{sat} = \frac{\Delta I_a}{1-\bar{r}r_s}\left[e^{-\tau/\mu} + \mathrm{T}(\mu)\right] \qquad (4)$$
The $\Delta I_a$ and $\Delta I_{sat}$ are the spatial variance in TOA radiance within an artificial light source for
aerosol and cloud free, and cloud free conditions, respectively. Similar to McHardy et al. (2015),
the spatial variance in radiance in this study is represented by the standard devation of radiance
within an artificial light source. Also, the diffuse transmittance, $T(\mu)$, is required. Following
Johnson et al. (2013), we estimated the ratio ($k$) between direct transmittance ($e^{-\tau/\mu}$) and total
transmittance using the 6S radiative transfer model (Vermote et al., 1997). This approach can also
be shown as Eq. 5:
$$k = e^{-\tau/\mu}/\left[e^{-\tau/\mu} + \mathrm{T}(\mu)\right] \qquad (5)$$
The look-up-table (LUT) values of $k$ were computed for the AOT ranges of 0-1.5 (with every 0.05 AOT
interval for AOT < 0.6 and for every 0.1 AOT interval for AOT of 0.6-0.1, and with two high AOT values
of 1.2 and 1.5), for three different aerosol types: dust, smoke, and pollutants. We also modified the 6S
model (Vermote et al., 1997) to account for the spectral response function of the VIIRS DNB band (e.g.,
Chen et al., 2017). No sea salt aerosol was included in the LUT for this study, as artificial light sources





considered in this study were inland with less probability of sea salt aerosol contamination. Still, sea salt
aerosol can be added in later studies. Thus, we can rewrite Eq. 4 as:

$\tau = \mu \ln \frac{\Delta I_a}{k \Delta I_{sat}(1 - \bar{r} r_s)}$           (6)


As suggested from Eq. 6, nighttime column optical thickness ($\tau$) can be estimated using spatial
variances of an artificial light source over aerosol- and cloud-free conditions. The $\overline{rr}_s$ term arises from
the reflectance between the aerosol and the surface layers. This term is small for dark surfaces or
low aerosol loading cases, but could be significant for thick aerosol plumes over bright surfaces,
such as dust aerosols over the desert. We assume this term is negligible for this study. Note that
$\tau$ values from Eq. 6 include AOT as well as scattering (Rayleigh) and absorption (e.g., oxygen A
band) optical depth from gas species. To derive nighttime AOTs, 6S radiative transfer calculations
(Vermote et al., 1997) were used, assuming a standard atmosphere, to compute and remove the
component due to molecular scattering.
**2.3 Data pre-processing steps**
The VIIRS data pre-processing for nighttime aerosol retrievals is implemented through two
steps. First, artificial light sources are identified. Second, the detected artificial light sources are
evaluated against a known city database and a detailed regional analysis is performed. This latter
step is necessary to eliminate any unwanted "false" artificial light sources such as cloud
contamination or lightning strikes.
In the first step, conducted on individual 'granules' (~90 second orbital subsets) or composites
of adjacent granules, artificial light sources are selected after cloud screening and quality assurance
procedures. Since VIIRS nighttime aerosol retrievals assume cloud free conditions, cloud-
contaminated pixels must be removed using the VIIRS cloud products. Note that the nighttime
VIIRS cloud mask is thermal infrared based, and has its limitations in detecting low clouds





(especially over land), and thus additional cloud screening methods are also implemented as
mentioned in a later section. A single granule of VIIRS DNB radiance data is 4064 by 768 pixels
while, for the same VIIRS granule, the VIIRS cloud product reports values at 2032 by 384 pixels.
Thus, the VIIRS cloud product is first oversampled and then used to screen the radiance data.
Following the cloud screening step, VIIRS DNB Quality Assurance (QA) flags are used to
eliminate pixels that either have missing or out-of-range data, exhibit saturation, or have bad
calibration quality. We require the solar zenith angle to be larger than 102° to eliminate solar
(including twilight) contamination. Upon cloud screening and QA checks, artificial light pixels
are detected using a threshold based method by examining the difference in radiance of a given
pixel to background pixels, as suggested in Johnson et al. (2013). Artificial light pixels are defined
as pixels having radiance values greater than 1.5 times that of the granule or multi-granule mean
cloud-free background radiances.
The implementation of the first pre-processing step is illustrated in Figs. 2a-2d. Figure 2a
shows VIIRS DNB radiance data over North America for Oct. 1, 2015. Figure 2b shows the same
data as Fig. 2a but with cloud screening (shown in gray) and QA steps applied. Data removed by
the day/night terminator (i.e., solar zenith angle < 102°) are shown in cyan, and pixels with QA
values indicating signal saturation are shown in yellow. Pixels in orange color in Fig. 2c are the
detected potential light sources on the granule scale. As shown in Fig. 2c, some cloud pixels may
still be misclassified as artificial light sources. To avoid such false detection, the detected artificial
light sources are further evaluated against a list of known cities for a given region as mentioned in
Section 2. This step is shown in Fig. 2d, where green colored pixels are artificial light sources
confirmed by the known city light source database. Here, only 200 arbitrarily selected cities in the



US were used, and thus some of the artificial light sources, although positively identified, were
not highlighted in green as they were not in the city list.
The granule or multi-granule mean cloud-free background radiances are used for detecting
artificial light sources in the first step, which may introduce an over- or under-detection of artificial
light sources. To refine this detection, a regionally based artificial light source detection step is
implemented. In this step, a bounding box is selected for each cloud-free city. The bounding
boxes are manually selected for 200 cities in the US and 8 cities in the Middle-East. Based on
experimenting, we found that most cities have a bounding box size of less than ±0.3°
latitude/longitude, except for large cities that have a population of ~quarter-million or more,
depending on countries. Thus, for the remaining 991 cities in the Middle-East and 2995 cities in
India, to simplify the process, a ±0.3° latitude/longitude region was picked as the bounding box.
The bounding boxes for large cities need to be manually selected in future studies.
Even if a city is partially included in a bounding box, or multiple cities reside within a bounding
box, retrievals can still be performed, since variances of detected artificial light sources are used
for aerosol retrials regardless of origins of those artificial light sources. The latitude and longitude
ranges of the bounding boxes for all cities used in the study are included in the attached city list
files. Similar steps as mentioned in the granule or multi-granule level detection scheme are
implemented here, but with the use of localized mean cloud-free background radiances. The
results from the regional detection is shown in Fig. 3. Figure 3a is the VIIRS nighttime image for
Sioux City, Iowa for April 13, 2015. The detected artificial light sources are shown in Fig. 3b,
where pixels with green color represent artificial light sources that are identified based on the local
detection scheme (the second step) while the pixels with orange color represent pixels identified



at the granule or multi-granule level (the first step) but fail on regional detection or outside the
bounding box.

Cloud contamination remains an issue in the above steps, as shown in Fig. 2c, owing to

limitations in the VIIRS infrared-based nighttime cloud mask. To further eliminate cities that are
partially covered by clouds, for a given artificial light source, nights with mean latitudes and
longitudes from detected light source pixels that are larger than 0.02° of the seasonally or yearly
mean geolocations are excluded. This process is based on the assumption that for a partially cloud
covered city, only a portion of the city is detected as artificial light source, and thus the mean
geolocations likely deviate from the multi-night composited mean geolocations. However, this
step may misidentify heavy aerosol plumes as cloud contaminated scenes. These nuances of city
light identification remain a topic of ongoing research, and for now remain as an outstanding
source of uncertainty in the current retrieval algorithm.

On each night and for each light source, the averaged radiance, its standard deviation, the lunar

fraction, viewing geometries, and the number of artificial light source pixels identified, are
reported as diagnostic information. To further avoid contamination from potential cloud / surface
contaminated pixels, or from pixels with erroneously high radiance values due to lightning flashes,
in the process of computing standard deviation the top 0.5% and bottom 10% of pixels are
excluded. Finally, this dataset is further used in the retrieval process.

**3. Results**
**3.1. Linkages between artificial lights and observing conditions**

As mentioned in Section 2, 200 cities within the US were arbitrarily chosen to examine the

properties of artificial light sources, as we expect less significant aerosol contaminations over the



US in comparison to other regions considered in this study. This analysis allows us to gain insight
on the natural variations of artificial light sources as a function of various observing parameters—
variations that will determine the inherent uncertainty of aerosol retrievals.
Cities have varying spatial light patterns, populations, and nighttime electricity usage, as well
as different surface conditions. To study the overall impacts of the observing conditions on
artificial light source patterns, the yearly mean radiance and standard deviation of the detected
light sources were computed for each city, regardless of observing conditions. Then, for each city
and for each night, the instantaneous radiance and standard deviation values were scaled based on
yearly mean values to derive a yearly mean normalized radiance (N_Radiance) and standard
deviation (N_$R_{std}$). This process was necessary to remove city-specific characteristics, making
feasible the comparison of artificial light source properties from different cities. Also, to remove
nights with cloud contamination or bad data, the yearly mean (N) and standard deviation (N_STD)
of the total number of light source pixels identified for a given artificial light source was computed.
Only nights with a number of detected light source pixels exceeding N - 0.1×N_STD were used in
the subsequent analysis.
Figure 4a shows the plot of Julian day versus normalized radiance using data from all 200
cities on all available nights, regardless of the observing conditions (with the exception of totally
cloudy scenes, as identified by the VIIRS cloud product, which were removed). As suggested
from Fig. 4a, nighttime artificial light sources vary as a function of Julian day. Higher radiance
values were found over the Northern Hemisphere winter season (Julian days greater than 300 or
less than 100, corresponding to the months of November through March of the following year),
compared to the Northern Hemisphere spring, summer, and fall seasons. In particular, during the
Northern Hemisphere winter season, high spikes of radiance values were clearly visible. The



increase in radiance values as well frequent high spikes in radiance values during the winter season
may be due in part to snow and ice reflectance (modifying the surface albedo, and hence the
multiple scatter between the atmosphere and surface as well as augmented lunar reflectance),
especially for high latitude regions. Thus, snow and ice removal steps are needed for nighttime
aerosol retrievals on both regional and global scales. Still, upon characterizing the snow/ice cover
from daytime observations, retrievals may still be possible over snow/ice contaminated regions for
future studies.
Also apparent in Fig. 4a is variation in the number of non-totally cloudy observations with
respect to Julian day. The minimum number of non-overcast observations that passed the QA
checks occurs during the months of June and July, likely due to saturation QA-flagged pixels
(colored in yellow in Fig. 2) reaching the furthest south during those two months. VIIRS DNB
QA checks also label a block of pixels adjacent to the day/night terminator as pixels with bad QA
(e.g., the yellow colored area in Fig. 2b). Thus, during June and July, a significant portion of
artificial light sources at high latitudes were removed from the analysis. These QA steps are
retained in the process, although relaxing these QA requirements may be an option for enhancing
data volume over high latitudes. An assessment of the uncertainties incurred by reducing the
conservative nature of the QA flag is a subject of future studies.
Figures 4c and 4e show that the yearly mean normalized radiance, N_Radiance, varies as a
function of lunar status, including the lunar fraction and lunar zenith angle. As the lunar fraction
increases, the N_Radiance increases, possibly due to the increase in reflected moon light. As lunar
zenith angle increases (i.e., the moon is less high in the sky), a decrease in the N_Radiance is
found, indicating a reduction in downward moon light as lunar zenith angle increases. An





interesting relationship between the N_Radiance and satellite zenith angle emerges in Fig. 4g. A

10-20% increase in N_Radiance is observed for an increase of satellite zenith angle from 0 to 60°.

Figures 4b,d,f,h show similar analyses as Figs. 4a,c,e,g but for $N\_R_{std}$. A similar relationship

between $N\_R_{std}$ and Julian day is also found, with larger $N\_R_{std}$ values found in winter and smaller

values found in the summer. Also, larger spikes of $N\_R_{std}$, possibly due to snow and ice

contamination, are found in the winter season, suggesting that careful ice and snow detection

methods are needed for processing VIIRS DNB data over high latitudes during the winter season.

Still, the increase in nighttime radiance and standard deviation of radiance may also be due to the

increase in artificial light usage at night during the winter months, and for this reason, seasonal or

monthly based $\Delta I_a$ values may be needed. In contrast to the normalized radiance, insignificant

changes in $N\_R_{std}$ were observed with the varying of either lunar fraction or lunar zenith angle,

indicating that lunar fraction or lunar zenith angle have less impact on nighttime aerosol retrievals

when considering $N\_R_{std}$.

$N\_R_{std}$ was found to be strongly dependent upon the satellite zenith angle, with values larger

than 1 observed at near 60° viewing zenith angle, likely due to the anisotropic behavior of artificial

light sources, as well as longer slant paths although the true reason remains unknown. To account

for this viewing zenith angle dependency, a correction factor c was introduced in Johnson et al.

(2013) in anticipation of this result. Based on Fig. 4h, the correction factor, c, specified as a

function of the satellite viewing zenith angle ($\theta$), was calculated using VIIRS DNB data from 2015

over the 200 selected cities:

$$c = 1.66 - 1.75 \times \cos(\theta) + 0.91 \times \cos(\theta)^2 \qquad (7)$$

Radiance and standard deviations values from this study were further divided by c to account for

the viewing angle dependency.




Figure 5a is a scatterplot of N_Radiance versus N_R$_{std}$. A strong linear relationship is shown
with a correlation of 0.92, suggesting that brighter artificial light sources are typically associated
with larger spatial variations in radiance. Figure 5b shows the relationship between N_R$_{std}$ and
AOT using a collocated VIIRS DNB and AERONET dataset. Only data from non-winter months
(April-October, 2015) were considered. Since nighttime AERONET data are not available, the
AERONET data used for the AOT comparisons in Fig. 5b are taken from the day immediately
prior and after the VIIRS nighttime observations, following the same collocation method as
described in Section 2. Figure 5b shows correlation between N_R$_{std}$ values and collocated
AERONET AOTs, and N_R$_{std}$ decreases as AOT increases. As such, Fig. 5b justifies the rationale
for retrieving nighttime AOT using spatial variations in artificial light sources.

**3.2 Parameter quantification for nighttime aerosol optical depth retrievals**
As shown in Eq. 6, to retrieve nighttime AOT using VIIRS DNB, $\Delta I_a$, $\Delta I_{sat}$, and k values must
be quantified. $\Delta I_{sat}$ is the standard deviation of an artificial light source under cloud-free
conditions, calculated directly from VIIRS DNB data. $\Delta I_a$ is the spatial standard deviation of the
same artificial light source but under aerosol and cloud-free conditions. The $\Delta I_a$ shall be derived
over nights with minimum aerosol contamination, or in principle, from nights with the highest
standard deviation of radiance (R$_{std}$) values. However, given that some of the highest R$_{std}$ values
may correspond to unscreened clouds or lightning, for a given year and for a given city we
computed the mean (R$_{std\_ave}$(30%)) and standard deviation (R$_{std\_std}$(30%)) of the 30% highest R$_{std}$
values. We then used the mean plus 2 times standard deviation of 30% highest R$_{std}$ values
(R$_{std\_ave}$(30%) + 2× R$_{std\_std}$(30%)) to represent the $\Delta I_a$ value. Assuming a normal data distribution,
two standard deviations above the mean R$_{std\_ave}$(30%) values should represent the top 1% of the





highest $R_{std}$ values of all data points—providing a way to compute the highest $R_{std}$ value while
simultaneously minimizing cloud and lightning contamination.   Artificial light sources are
excluded if the ratio of $R_{std\_std}(30\%)$ to $R_{std\_ave}(30\%)$ is above 15%.  Those artificial light sources
with larger variations in peak $R_{std}$ values are likely to be associated with cities that have less stable
artificial light signals.  Over the US, because of the concerns for ice and snow contamination as
mentioned in Sect. 3.1, only data from non-winter months (April-October, 2015) were used.  For
the India and Middle East regions, snow and ice contamination is likely insignificant and thus data
from all months in 2015 were used.

As mentioned in Sect. 2.2, k values are computed using a LUT (pre-computed using the 6S

radiative transfer model) for dust, smoke, and pollutant aerosols.  For simplicity, we assumed the
US, Middle East, and India regions were dominated by pollutant, dust, and smoke aerosols,
respectively.  In future applications, k values (related to aerosol type) shall either be evaluated on
a regional basis, following Remer et al. (2005), or derived directly from VIIRS as mentioned in a
later section.

Cloud contamination is a long-standing challenge to passive-based satellite aerosol research

(e.g., Zhang et al., 2005).  In this study, the VIIRS cloud product (VCCLO) was used for cloud
clearing of the observed VIIRS DNB scenes.  However, only VIIRS Infrared channels are applied
for cloud detection at night (Godin and Vicente, 2015).  Thus, it is possible that low level clouds,
unseen by the VIIRS nighttime cloud mask, may still be present in the "cloud-cleared" scenes.  To
further exclude potential cloud contaminated artificial light sources, we have implemented
additional quality control steps.  First, it is noted that in the presence of low clouds certain artificial
light source patterns may appear differently from clear-sky conditions.  Thus, only nights with
mean geolocations of the detected artificial light sources that are within 0.02° of multi-night clear



sky means are used. This approach, however, will introduce issues for regions with persistent
cloud or thick aerosol plume coverage, such as the Uttar Pradesh state of India, which is mentioned
later.
It was noted in Sect. 3.1 that the radiance and standard deviation of radiance are strongly
correlated. As such, for each city and for each year, a regression relationship between radiance
and standard deviation of radiance values was constructed by calculating mean and standard
deviation of $R_{std}$ for a given radiance range. For a given range of radiance values, $R_{std}$ values that
were two standard deviations above the mean $R_{std}$ for that range were discarded as noisy data.
After removing these noisy points, the same procedures were repeated to compute the regression
between radiance and $R_{std}$ values for each city. The overall mean of $R_{std}$ ($R_{std\_mean}$) for the given
artificial light source was also computed. Data were removed if the $R_{std}$ value was above the
estimated $R_{std}$ based on radiance values using the above discussed regression plus 0.5 times
$R_{std\_mean}$. This step was taken to further remove cloud contaminated data, but may also remove
scenes with thick aerosol plumes.

**3.3 Regional retrievals**

One of the goals of this study is to apply the proposed algorithm on a regional scale. A full
retrieval and evaluation, using modified schemes as identified from this paper, will be conducted
in follow-up research. Here, we present preliminary results conducted on a regional scale for three
selected regions in 2015: the US, Middle East, and India. As mentioned previously, only non-
winter months were used (April-October) for the US region due to concerns of snow and ice
contamination, while all months were included for the other two regions.





Figure 6a shows the comparison between retrieved nighttime AOTs from VIIRS DNB and
collocated daytime AERONET AOTs (0.675 µm) for the selected 200 cities for 2015. Here VIIRS
DNB AOTs are retrieved without using the k (diffuse transmittance) correction term mentioned in
Sections 2 and 3.2. A total of 368 collocated points are found with a correlation of 0.59. Figure
6b shows the collocated CALIOP and VIIRS nighttime AOTs, again using the retrievals without
correcting for the diffuse transmittance term. A correlation of 0.47 was found between CALIOP
AOT (interpolated to 0.700 µm) and VIIRS nighttime AOT.
Figures 6c and 6d show retrieval comparisons similar to Figs. 6a and 6b, but revised to include
the $k$ (diffuse transmittance) correction term. An over correction was found as a higher than 1
slope between VIIRS and daytime AERONET AOTs, indicating that the correction for diffuse
transmittance may be less important for low aerosol loading cases. The daytime AERONET AOT
may not be a fair representation of nighttime AOTs in all cases. Large uncertainties exist in
CALIOP extinctions and AOTs as well, due to necessary assumptions of the lidar ratios made in
the retrieval process (e.g., Omar et al., 2013). Therefore, significant uncertainties exist in both the
AERONET and CALIOP validation sources. Still, this can be improved with the use of nighttime
lunar photometry data that is in development from the AERONET group (e.g., Berkoff et al., 2011;
Barreto et al., 2013).
Figures 7a and 7b show scatter plots of VIIRS DNB AOTs vs. daytime AERONET and
nighttime CALIOP AOTs, respectively, for the Middle East for 2015, using retrievals without $k$.
A total of 999 cities were included in the study, and 368 cities were excluded for not passing the
stable light source check (or $R_{std\_std}(30\%)$ / $R_{std\_ave}(30\%) < 15\%$) or not having 3 or more nights
that passed the various checks as mentioned in previous sections (both criteria are referred as the
stable light source requirement). A correlation of 0.64 and 0.46 was found between VIIRS and



AERONET and CALIOP AOTs, respectively. However, a low bias was clearly present in both
comparisons. Figures 7c and 7d show the VIIRS nighttime AOTs versus AERONET (day) and
CALIOP (night) AOTs with $k$ included. Similar correlations are found, yet the low bias is largely
corrected.

A similar study was conducted for India. Here we separated cities in India inside and outside

of the Uttar Pradesh (UP) state (retrieval for the UP state is discussed later). Of a total of 2573
cities outside of the UP state, 1810 cities were found to satisfy the stable light source requirement.
Again, Figs. 8a and 8b are for VIIRS nighttime AOTs versus AERONET adjacent daytime and
CALIOP nighttime AOTs without k correction and Figs. 8c and 8d are the plots with the diffuse
transmittance ($k$) correction term included, for cities that were outside the UP state. In all four
cases, correlations of around 0.5-0.6 were found, indicating the developed algorithm has
reasonable skill in tracking nighttime AOTs. A low bias occurred when $k$ was not included. When
$k$ was included, a near 1-to-1 agreement is found in both Figs. 8c and 8d. This exercise reinforces
the notion that there is indeed a need to account for diffuse transmittance.

Figures 9a and 9b compares VIIRS, AERONET, and CALIOP reported AOTs for cities within

the UP state of India. Of a total of 421 cities, 326 passed the stable city light requirement.
However, a low correlation was found between VIIRS nighttime and daytime AERONET AOTs.
This result is not surprising, as thick aerosol plumes cover this region most times of the year, and
thus the derived cloud and aerosol free sky standard deviation of the artificial light sources (the
$\Delta I_a$ values) are not always representative of true aerosol-free cases. Therefore, a longer study
period, or careful by-hand analysis, may be needed for deriving $\Delta I_a$ values for regions that are
known to have persistent thick aerosol plume coverage.



Ideally, the retrievals at each light source location should be gridded and averaged to further
increase the signal-to-noise ratio. We have tested this concept by averaging retrievals shown in
Figs. 6b, 7b, and 8b into a 1° x 1° (Latitude/Longitude) averaged dataset. Artificial light sources
that have less than 20 valid nights in a year were excluded to provide statistically robust estimates
of $\Delta I_a$. Comparisons of 1° x 1° (Latitude/Longitude) averaged VIIRS DNB AOT retrievals with
daytime AERONET data and nighttime CALIOP AOTs are shown in Figs. 6e (6f), 7e (7f), and 8e
(8f) for the US, Middle East, and India regions, respectively. Increases in correlations were found
between VIIRS and AERONET AOTs for the India regions. Marginal changes in correlations,
however, occurred between VIIRS and CALIOP AOTs. Although neither daytime AERONET
nor nighttime CALIOP AOTs can be considered as the "ground truth" for nighttime AOTs, these
results suggest that the newly developed method has skills in retrieving nighttime AOTs over both
dark and bright surfaces.
Figure 10 shows nighttime AOT retrievals over India for Jan. 12 and 16 of 2015, with the
retrievals from the UP state of India removed. Figures 10a and 10b show true color imagery from
Terra MODIS for Jan. 12 and 16, 2015 (obtained from the NASA Worldview through the
following site: https://worldview.earthdata.nasa.gov/). Figure 10c and 10d show the nighttime
images of VIIRS DNB radiance for Jan. 12 and 16, 2015. Over-plotted on Figs. 10a and 10b are
retrieved VIIRS nighttime AOTs, with blue, green, orange, and red representing AOT ranges of 0-
0.2, 0.2-0.4, 0.4-0.6, and above 0.6, respectively, using gridded data same as used for Figs. 9e-f.
Shown in Fig. 10a, on Jan. 12, the west portion of India was relatively aerosol-free, but a heavy
aerosol plume is visible around the east coast of India. Similarly, AOTs lower than 0.2 were
detected over western India but AOTs larger than 0.6 were found over eastern India. On Jan.16,
as indicated from the MODIS daytime image, a thick plume covered the western portion of India,





also seen in Fig. 10d via retrieved AOTs above 0.6. Also, the northeast portion of India was

relaltively aerosol-free as indicated from both MODIS true color imagery (Fig. 10b) and VIIRS

nighttime AOT retrievals (Fig. 10d).

Based on Figs. 10c and 10d, there were many artificial light sources not used in the retrieval.

Those sources were excluded by various quality-control checks of the study due to such reasons

as potential cloud contamination, light source instability, or insufficient valid data in a year. It is

very likely that some valid data will be removed in this conservative filtering process. New

methods must be developed to restore valid data. Some ideas to this effect are presented in the

section to follow.

The diffuse correction term, $k$, was shown to be an important factor in reducing bias in these

retrievals. We compared the k corrections estimated using the 6S model (Vermote et al., 1997) as

well as those empirically derived from this study. By assuming CALIOP nighttime AOTs as the

"true" AOTs, and using VIIRS AOTs as shown in Figs. 7b and 8b as inputs, the k correction term

could be inferred using Eq. 6. Figure 11a shows the derived $k$ values vs. CALIOP nighttime AOT

for the Middle East region. Over-plotted are the $k$ values estimated from the 6S model (Vermote

et al., 1997). The two patterns show some agreement, as both the modeled and the empirically

derived $k$ values are near or above 1 for CALIOP AOTs of 0.0, and below 0.5 when CALIOP

AOTs of ~1. This behavior indicates that the 6S-modeled $k$ correction may provide a reasonable

first-order estimate for dust aerosols in this region. Figure 11b shows a similar plot as Fig. 11a

but for the India region. A larger data spread was found between the empirically derived and

modeled $k$ values assuming smoke aerosols, although the overall patterns were similar. One of the

possible reasons for the disparity is that unlike the Middle East region, where dust aerosols



dominate, the India region is subject to many other aerosol species including dust and pollutants,
occurring across different regions and varying with season.

**545 3.5 Limitations and possible improvements**

Although showing some skill, the retrieval algorithm examined in this study has its limitations.
First, most retrievals are limited to AOTs less than 1.5. This is because scenes with heavy aerosol
plumes can either be misclassified as clouds by the VIIRS cloud product, or removed during the
additional cloud screening steps introduced in this study. For heavy aerosol plumes, much larger
areas could be detected as "light sources" due to enhanced diffuse radiation (e.g., Figure 11), and
have different mean geolocations than low aerosol loadings and cloud free nights, and thus would
be removed due to the geolocation checks as mentioned in Section 3.2. A data loss, especially for
heavy aerosol cases, is experienced in this study due to those stringent data screening steps. Also,
for the purpose of avoiding cloud or lightning contamination in this study, $\Delta I_a$ values were not
derived from nights with the highest radiance or standard deviation of radiance values. Doing so
creates a problem for regions having frequent heavy aerosol plume loading, such as the UP state
of India.
Both issues mentioned above may be mitigated by constructing a prescribed city pattern for
each light source based on a multi-night composite from cloud free and low aerosol loading
conditions. In that case, light source pixels from the exact same location would be used each night
to reduce data loss, especially for nights with heavy aerosol plumes. In constructing the predefined
city pattern, $\Delta I_a$ values may also be derived. The construction of a prescribed city pattern will be
attempted in a future study.





Even after vigorous attempts at cloud screening, there remains some cloud contamination. Such conditions may account in Figs. 6-8 for high VIIRS AOT but low CALIOP or AERONET AOT cases, although both daytime AERONET data and CALIOP data have their own issues for representing nighttime aerosol optical depth, as discussed. More advanced cloud screening methods are needed to improve the screening-out of residual clouds. In addition, snow and ice cover pose challenges for this study, and new methods need to be developed to account for snow / ice coverage and allow for attempts at nighttime AOT retrievals over those scenes.

Even the algorithm as presented shows skill in retrieving nighttime AOT. Given that there are hundreds of thousands of cities and towns across the world that could serve as sources for this algorithm, the composite of retrievals from artificial light sources may provide a tractable means to attaining regional to global description of nighttime aerosol conditions, on both moonlit and moon-free nights, and over both dark and bright land surfaces. Considering the current glaring nocturnal gap in AOT, the current results show promise for providing closure and thereby enabling cloud/aerosol process studies and improved parameterizations for weather and climate modeling.

## 4    Conclusions and Implications

In this study, based on Visible/Infrared Imager/Radiometer Suite (VIIRS) Day/Night band (DNB) data from 2015, we examined the characteristics of artificial light sources for selected cities in the US, India, and the Middle East regions. Our findings point toward the following key conclusions:

1. Radiance from artificial light sources is a function of time of year, lunar illumination and geometry, and viewing geometry. Larger radiance values and spikes in radiance values can occur during the winter season, possibly related to snow and ice cover, indicating the





need for careful snow and ice detection for nighttime retrievals using VIIRS data. The normalized radiance increases with lunar fraction, and decreases with increasing lunar zenith angle—as these parameters are tied to the magnitude of downwelling moonlight.

2. The normalized standard deviation of artificial light source radiance is a function of time of year and similar to normalized radiance, exhibit spikes during the winter season. However, no significant relationship was found between the normalized standard deviation of radiance and lunar characteristics, including lunar fraction and lunar zenith angle. This finding suggests that the standard deviation of radiance, as opposed to the normalized value of radiance, is a potentially more robust parameter for nighttime aerosol retrievals using VIIRS DNB data.

3. Both the normalized radiance and the normalized standard deviation of radiance are a strong function of satellite viewing angle, with larger normalized radiance and the normalized standard deviation of radiance values occurring at higher satellite viewing angles. As anticipated by past research, this viewing angle dependency must be accounted for in VIIRS DNB nighttime aerosol retrievals based on artificial light sources.

4. Preliminary evaluations over the US for 200 selected cities, over the Middle East for 999 cities/towns, and over India for 2995 cities/towns (excluding the Uttar Pradesh State of India) show reasonable agreements between VIIRS nighttime aerosol optical thickness (AOT) and values estimated by adjacent-daytime AErosol RObotic NETwork (AERONET) and nighttime Cloud-Aerosol Lidar with Orthogonal Polarization (CALIOP). This finding suggests that the use of artificial light sources holds the potential of being viable for regional as well as global nighttime aerosol retrievals.



5. Poor correlation was found between VIIRS nighttime AOTs and daytime AERONET AOTs for the Uttar Pradesh state in India. This region is frequently covered by thick aerosol plumes, and this may introduce a difficulty in constructing cloud and aerosol free night characteristics of artificial light sources ($\Delta I_a$) for the retrieval process. Based on this finding, we conclude that detailed analysis, and perhaps by-hand selection of non-turbid baseline conditions, is needed for estimating $\Delta I_a$ values in regions of climatologically high and persistent turbidity.

6. In contrast to McHardy et al. (2015), the need for a diffuse correction in the nighttime aerosol retrieval process was found to indeed be important for regions with heavy aerosol loadings. This study further suggests radiative transfer model based estimations of the diffuse correction term compare reasonably well with empirically derived values over the Middle East where the dominant aerosol type is dust. However, in cases such as the India region, where several aerosol types may be expected during a year, a larger data spread was found, and specification of the diffuse correction term requires additional study.

Despite the advances made here, there remain many limitations to the current algorithm. For example, snow, ice, and cloud contamination can significantly affect the retrieved AOTs. Advanced procedures for snow, ice, and cloud removal are needed, with a full evaluation for the potential impacts. Also, high aerosol loading may be screened out due to misclassification of thick aerosol plumes as clouds. A pattern-based artificial light source method will be examined in a future study as one approach to mitigating this issue. Despite these known issues, these low-light studies forge a promising new pathway toward providing nighttime aerosol optical property information on the spatial and temporal time scales of value to the significant needs of the aerosol modeling community in terms of regional to global nighttime aerosol property information.



**Acknowledgments:**

The support of the Office of Naval Research under grant N00014-16-1-2040 and the NOAA JPSS

Program Office are gratefully acknowledged. S. L. Jaker was partially supported by the NASA

Grant of NNX17AG52G and NSF project IIA-1355466. The global city base used in this study is

a free open source dataset. "This product includes data created by MaxMind, available from

http://www.maxmind.com/".




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





**Figure Captions**

**Figure 1**. Spatial distribution of (a) 200 cities over the US, b). 999 cities over the Middle East, and c). 2995 cities over India, used in this study. Red dots show cities/towns from the Uttar Pradesh (UP) state of India—a region of climatologically high aerosol loading.

**Figure 2.** (a) VIIRS DNB contrast-enhanced imagery centered over North America from the VIIRS DNB for October, 1 2015. (b) Same as (a), but with cloud screening and quality assurance steps applied for cloudy (grey), saturated pixels (yellow), and solar zenith angles < 102° (cyan). (c) Similar to (b), but with artificial light sources identified through a granule level detection (orange). (d) Similar to (c) but showing artificial light sources cross checked with a known city database and through a regional level detection (green).

**Figure 3.** (a) VIIRS nighttime imagery on April 13, 2015 over Sioux City, Iowa, US. b) Similar to (a) but showing detected artificial light sources using data within ±0.3° (Latitude/Longitude) of the city center (green). Orange colors show the detected artifical light sources through a granule level detection. Only green pixels are utilized for aerosol retrievals.

**Figure 4**. (a), (c), (e), and (g) show the normalized radiance of artificial light sources (200 selected cities over the US, for 2015) as functions of Julian day, lunar fraction, lunar zenith angle, and satellite zenith angle, respectively. (b), (d), (f), and (h) show the corresponding normalized standard deviation of radiance for artificial light sources.



**Figure 5**. (a) Normalized radiance versus normalized standard deviation of radiance for 200 cities over the US for 2015. (b) The normalized standard deviation of radiance as a function of adjacent daytime AERONET AOT (0.675 µm).

**Figure 6**. (a) Scatter plot of VIIRS nighttime AOT versus adjacent daytime AERONET AOT (0.675 µm) for 200 selected cities over the US for 2015. No diffuse correction is applied. b) Similar to (a) but for using nighttime CALIOP AOT (0.7 µm). (c) and (d)) Similar (a) and (b) but with the diffuse correction implemented. (e) and (f): Similar to Figs. 6c and 6d but for gridded VIIRS data (averaged into 1° × 1° Latitude/Longitude grids). Artificial light sources with fewer than 20 nights that passed various cloud screening and QA checks are excluded.

**Figure 7**. Similar to Fig. 6 but for 999 cities over the Middle East for 2015.

**Figure 8**. Similar to Fig. 7 but for the India region for 2015. Artificial light sources from the Uttar Pradesh State of India are excluded.

**Figure 9**. (a) Scatter plot of VIIRS nighttime AOT versus adjacent day time AERONET AOT (0.675 µm) over the Uttar Pradesh State of India for 2015. Diffuse correction is applied. (b): similar to (a), but for using nighttime CALIOP AOT (0.7 µm).

**Figure 10**. Terra MODIS true color imagery (NASA Worldview) for Jan. 12, 2015 over India. (b): Similar as (a) but for Jan. 16, 2015. (c): VIIRS nighttime imagery on Jan. 12, 2015. Over plotted are VIIRS nighttime AOT retrievals in 1°×1° (Latitude/Longitude) grid format. Blue,



green, orange, and red colors represent AOT ranges of 0-0.2, 0.2-0.4, 0.4-0.6 and > 0.6,
respectively. (d) similar to (c) but for Jan. 16, 2015.

**Figure 11**. (a) Empirically derived (using data from Fig. 7d) and 6S model estimated diffuse
correction terms for the Middle East for 2015. (b): Similar to Fig. 10a but for the India region for
2015 (using data from Fig. 8d).



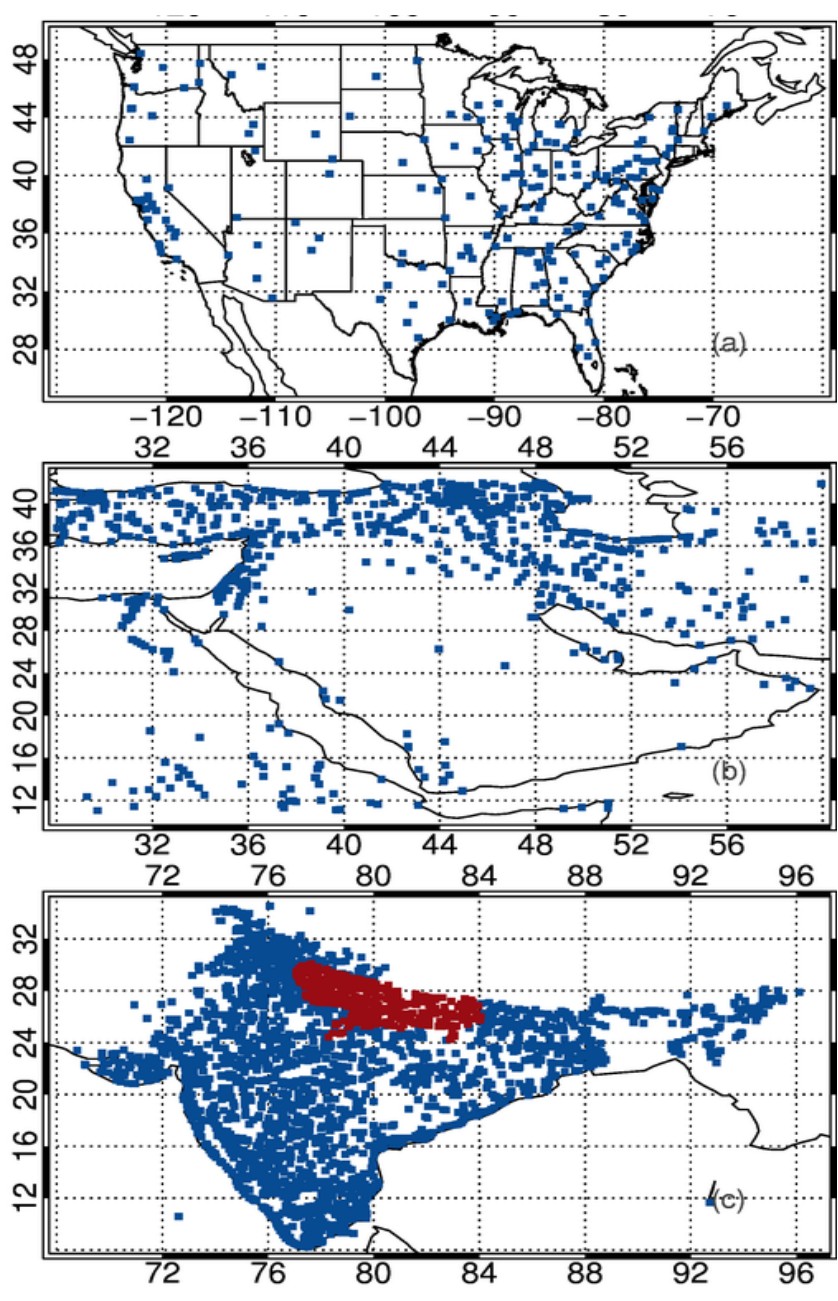


**Figure 1.** Spatial distribution of (a) 200 cities over the US, b). 999 cities over the Middle East,
and c 2995 cities over India, used in this study. Red dots show cities/towns from the Uttar
Pradesh (UP) state of India—a region of climatologically high aerosol loading.





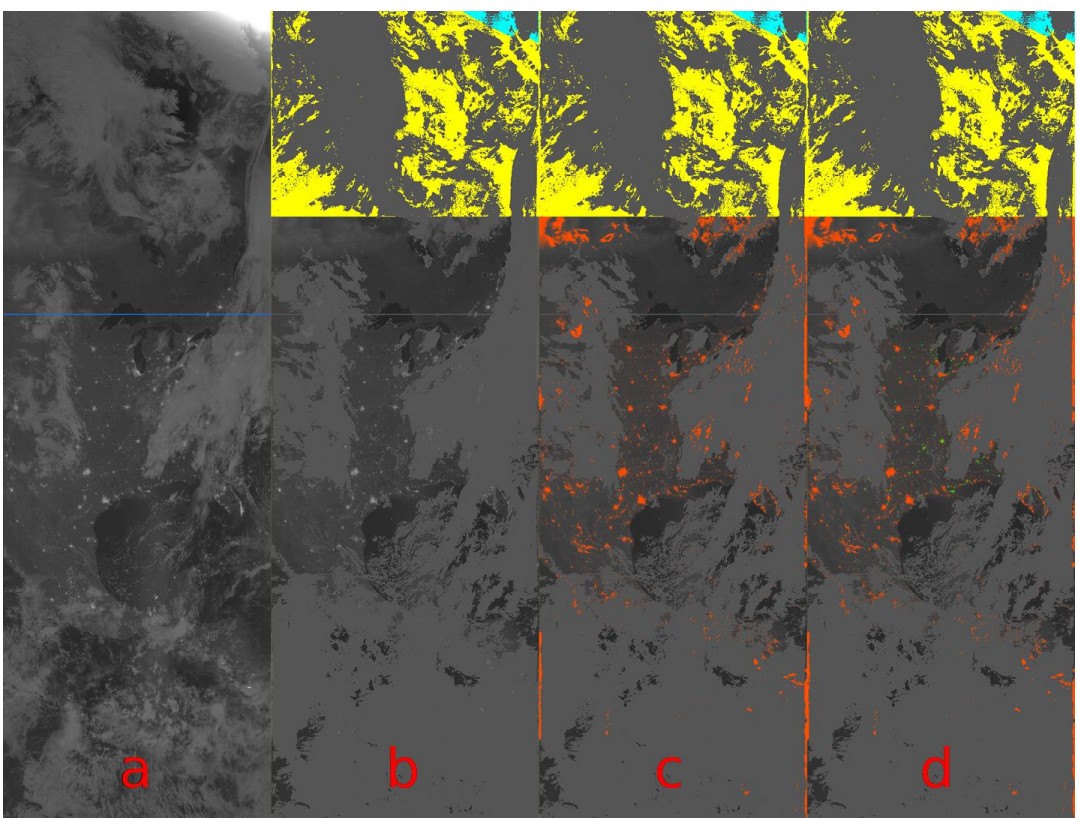

**Figure 2.** (a) VIIRS DNB contrast-enhanced imagery centered over North America from the VIIRS DNB for October, 1 2015. (b) Same as (a), but with cloud screening and quality assurance steps applied for cloudy (grey), saturated pixels (yellow), and solar zenith angle s< 102° (cyan). (c) Similar to (b), but with artificial light sources identified through a granule level detection (orange). (d) Similar to (c) but showing artificial light sources cross checked with a known city database and through a regional level detection (green).



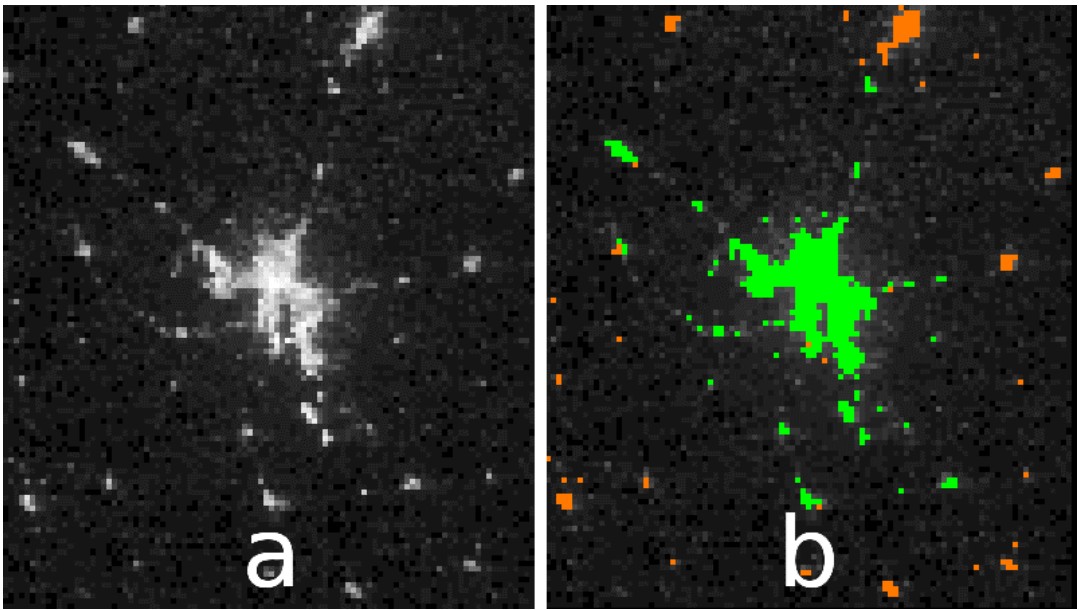

768

769

**Figure 3.** (a) VIIRS nighttime imagery on April 13, 2015 over Sioux City, Iowa, US. b) Similar to (a) but showing detected artificial light sources using data within ±0.3° (Latitude/Longitdue) of the city center (green). Orange colors show the detected artifical light sources through a granule level detection. Only green pixels are utilized for aerosol retrievals.










Figure 4. (a), (c), (e), and (g) show the normalized radiance of artificial light sources (200 selected cities over the US, for 2015) as functions of Julian day, lunar fraction, lunar zenith angle, and satellite zenith angle, respectively. (b), (d), (f), and (h) show the corresponding normalized standard deviation of radiance for artificial light sources.



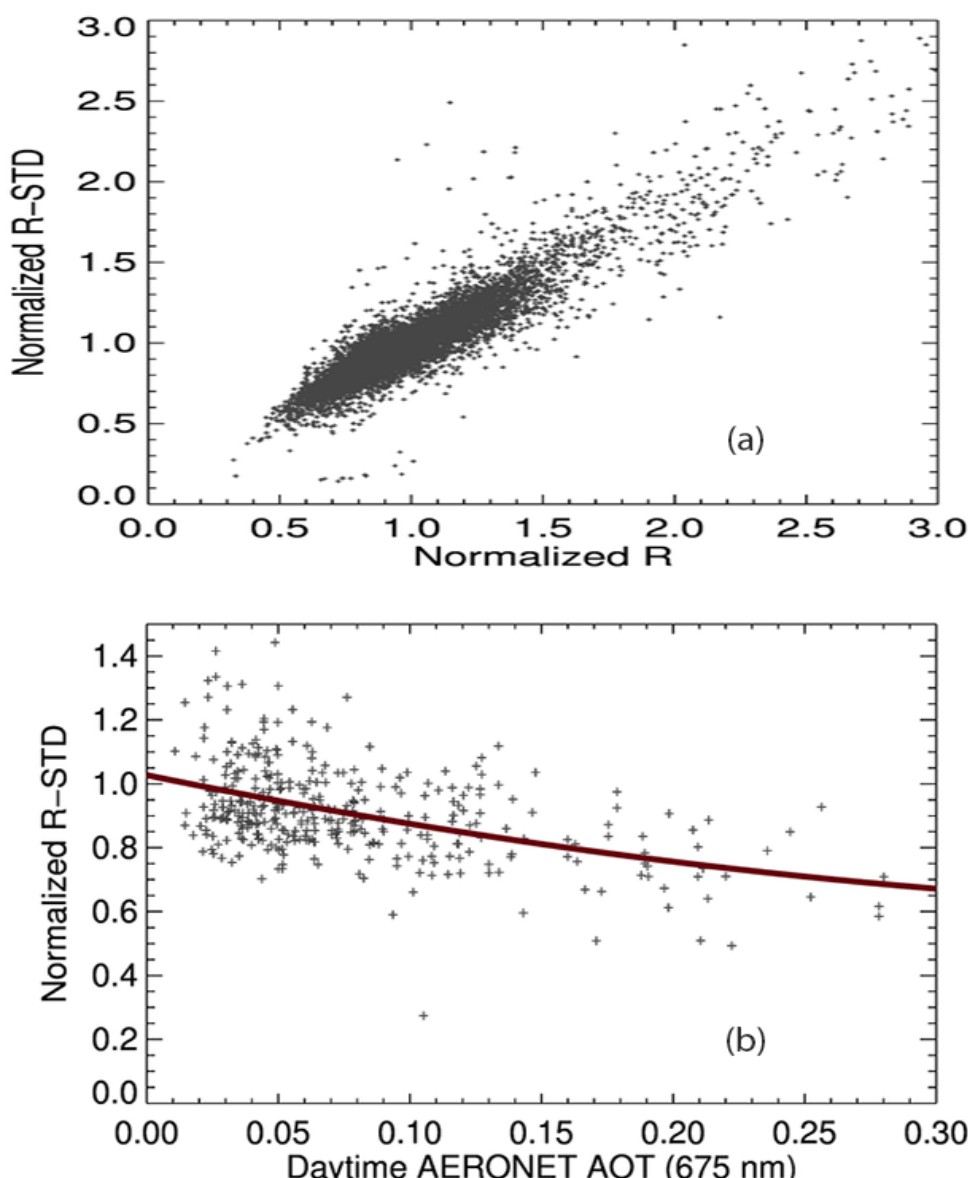

784

**Figure 5**. (a) Normalized radiance versus normalized standard deviation of radiance for 200 cities
over the US for 2015. (b) The normalized standard deviation of radiance as a function of adjacent
daytime AERONET AOT (0.675 μm).







**Figure 6**. (a) Scatter plot of VIIRS nighttime AOT versus adjacent daytime AERONET AOT
(0.675 μm) for 200 selected cities over the US for 2015. No diffuse correction is applied. b)
Similar to (a) but for using nighttime CALIOP AOT (0.7 μm). (c) and (d)) Similar (a) and (b) but
with the diffuse correction implemented. (e) and (f): Similar to Figs. 6c and 6d but for gridded
VIIRS data (averaged into 1° × 1° Latitude/Longitude grids). Artificial light sources with fewer
than 20 nights that passed various cloud screening and QA checks are excluded.



**Figure 7**. Similar to Fig. 6 but for 999 cities over the Middle East for 2015.






**Figure 8.** Similar to Fig. 7 but for the India region for 2015. Artificial light sources from the
Uttar Pradesh State of India are excluded.







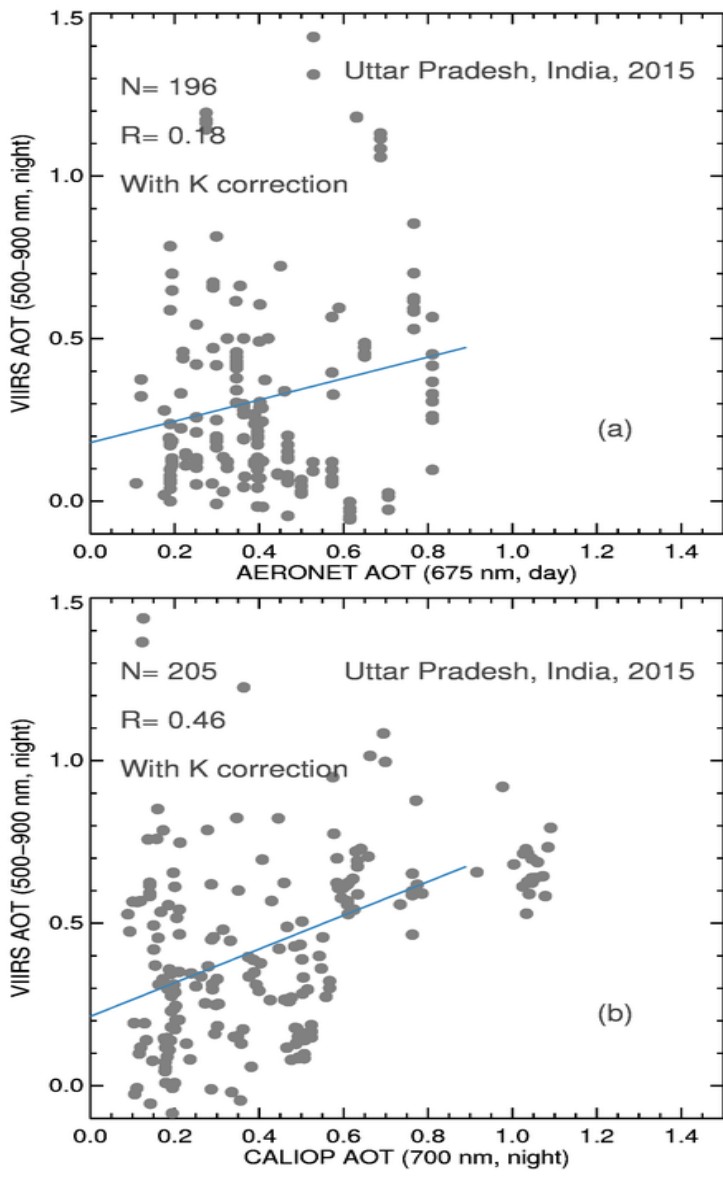

**Figure 9**.  (a) Scatter plot of VIIRS nighttime AOT versus adjacent day time AERONET AOT
(0.675 µm) over the Uttar Pradesh State of India for 2015.  Diffuse correction is applied. (b):
similar to (a), but for using nighttime CALIOP AOT (0.7 µm).

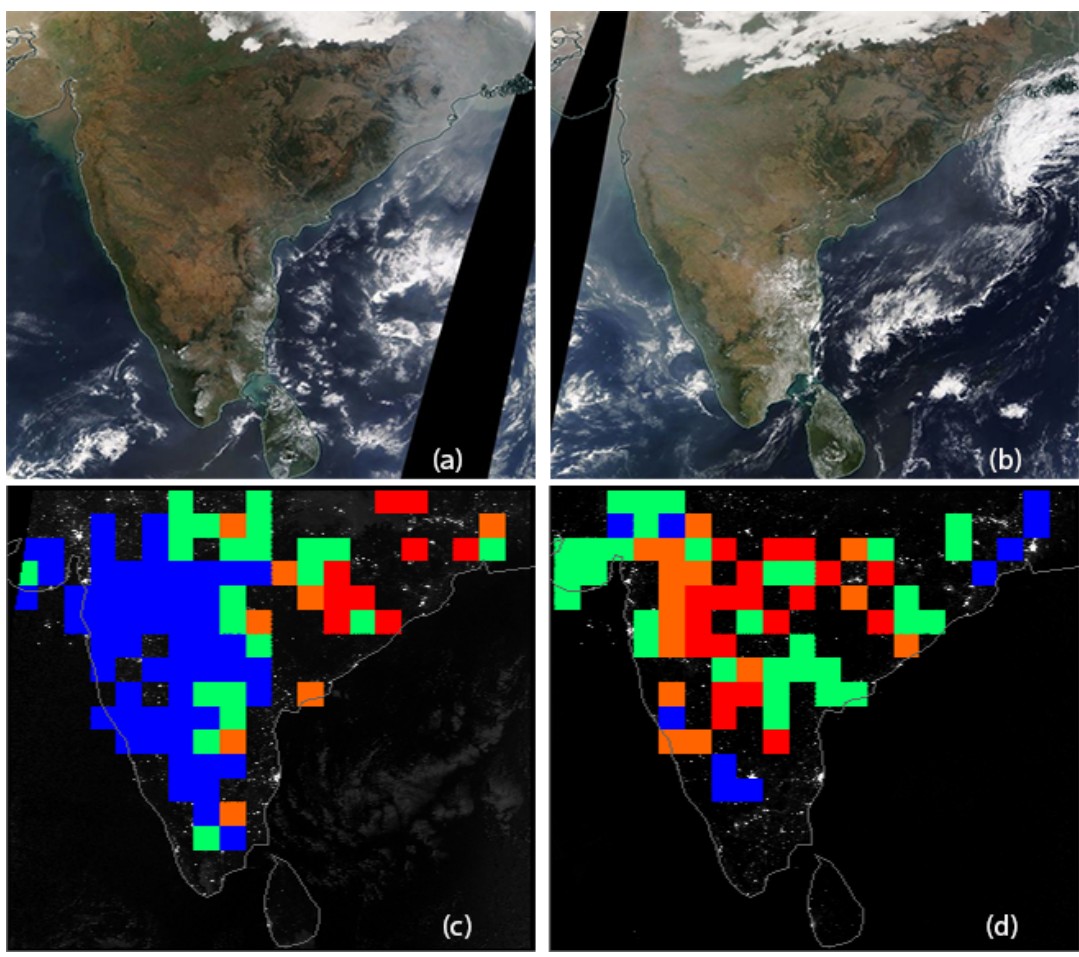

**Figure 10**. Terra MODIS true color imagery (NASA Worldview) for Jan. 12, 2015 over India.
(b): Similar as (a) but for Jan. 16, 2015. (c): VIIRS nighttime imagery on Jan. 12, 2015. Over
plotted are VIIRS nighttime AOT retrievals in 1°×1° (Latitude/Longitude) grid format. Blue,
green, orange, and red colors represent AOT ranges of 0-0.2, 0.2-0.4, 0.4-0.6 and > 0.6,
respectively. (d) similar to (c) but for Jan. 16, 2015.






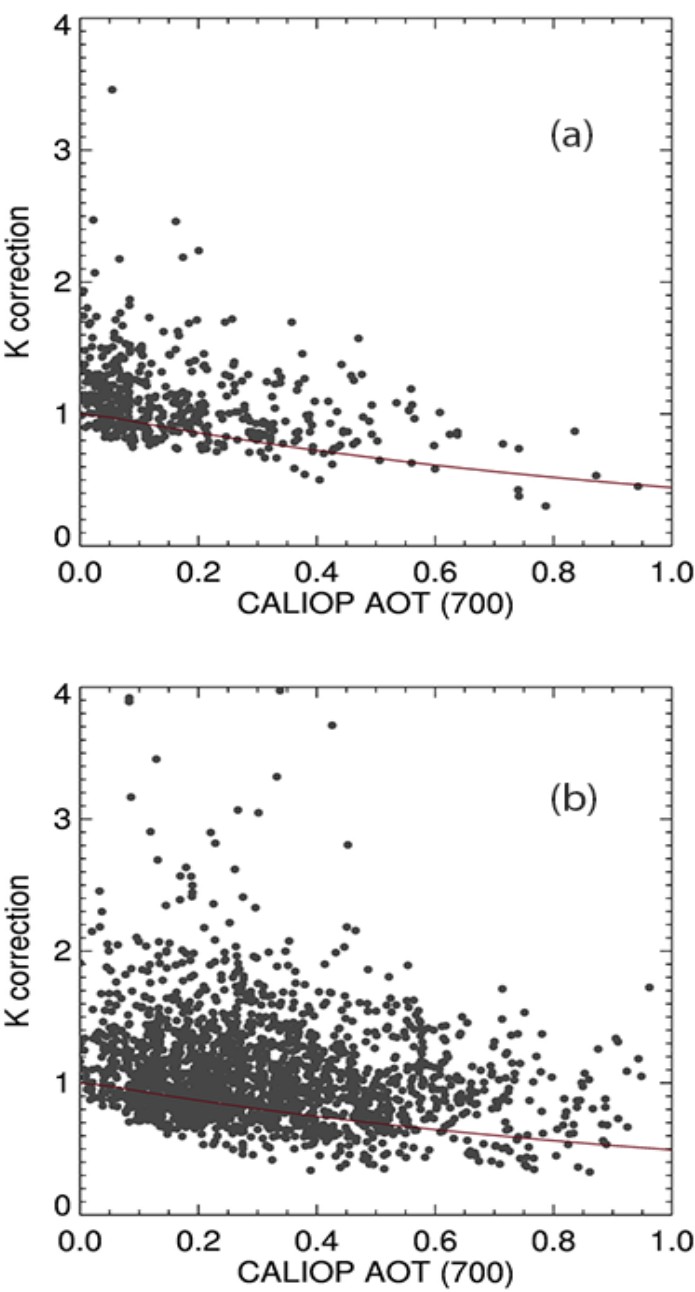


**Figure 11**. (a) Empirically derived (using data from Fig. 7d) and 6S model estimated diffuse
correction terms for the Middle East for 2015. (b): Similar to Fig. 10a but for the India region for
2015 (using data from Fig. 8d).