# Peer review of "Characterization and application of artificial light sources for nighttime aerosol optical depth retrievals using the VIIRS Day/Night Band"

_Atmospheric Measurement Techniques, 2018_

## Referee Comment (RC1) · Anonymous Referee #1 · 22 Jan 2019

This is the third paper from this group investigating the possibilities of deriving aerosol optical thickness at night from the VIIRS DNB. Interestingly I also reviewed the first paper in this series from 2013. I did not review or read the second paper in the series (McHardy et al., 2015) until I accepted this review. The group is making progress, but progress is slow. I would have hoped that they would have offered more than an incremental improvement to their 2015 paper, which is what we have here, and for awhile I was trying to determine whether this increment was sufficiently novel to warrant publication.

In this installment the authors implement a retrieval developed in the 2015 paper that

uses the blurring of the spatial variability of light emitting city pixels at night to derive aerosol optical thickness (AOT). This is different from the paper that I reviewed in 2013 that used the contrast between the lit cities and the surrounding dark countryside. The idea of linking AOT to reduced sharpness of the image is old, and I am thinking of Tanré and Legrand (1991). The goal of a night time AOT from an imager is fundamentally ambitious. Everything from cloud clearing to the instability of artificial lights to the impossibility of collocating retrievals with current AERONET observations is working against you. Yet, the pay off is extremely rich, and as an aerosol product user, I am very happy that this group has not given up hope.

The manuscript as it stands now could use some scrubbing to make it easier to understand, and I will make some recommendations. In some places the authors have made things complicated when simplicity should suffice. In particular the derivation of the cloud/aerosol free spatial variability measure for each target is complicated. It would also be helpful to start to quantify AOT accuracy using this method. Previous papers made that attempt, but here there was nothing.

Overall the technique and implementation are sound, the presentation is sufficiently clean and the results sufficiently novel for publication. Therefore, I recommend publication after revision.

Tanré, D., & Legrand, M. (1991). On the satellite retrieval of Saharan dust optical thickness over land: Two different approaches. Journal of Geophysical Research: Atmospheres, 96(D3), 5221-5227.

Point by point comments and recommendations.

Introduction. There are a few other groups besides this one that have attempted similar aerosol retrievals using the DNB, or investigated the possibility theoretically. In particular I recommend referencing Wang et al. (2016), and possibly Choo and Jeong (2016), though the latter is very local in its scope. Wang et al. (2016) put to rest the importance of variable water vapor in the method. If they hadn't, I would be worried about

that factor.

Wang, J., Aegerter, C., Xu, X., & Szykman, J. J. (2016). Potential application of VI-IRS Day/Night Band for monitoring nighttime surface PM2. 5 air quality from space. Atmospheric Environment, 124, 55-63.

Choo, G. H., & Jeong, M. J. (2016). Estimation of nighttime aerosol optical thickness from Suomi-NPP DNB observations over small cities in Korea. Korean Journal of Remote Sensing, 32(2), 73-86.

Lines 88-95. I found the use of present tense "develops" (line 88) and "proposes" (line 90) a little odd. Those studies are done and published. I would have used "developed" and "proposed", but I'm not sure this is required.

2.1 Data sets

What is the time span or dates of the study? Is it one year of data or multiple years? I don't see this stated anywhere.

Likewise no where is it stated that this is specifically Suomi-NPP VIIRS, because there are now more than one VIIRS flying. There's NOAA 20, and over the years there will be more.

The specific data sets look to me that they are IDPS data sets. IDPS processing and archiving of data sets has stopped or will stop very soon to be replace by EPS data sets and processing. I think this should be clarified in the text.

Line 172 "attachment". First should this be "attachment" or "supplements"? Note that I was unable to down load or to see any of the supplementary files. This is likely a problem between me and the journal, not with the authors, but this should be tested.

2.2 Retrieval methods

Line 182. "optical depth". Elsewhere the authors use the term "optical thickness". It would be good to be consistent.

Line 219. I believe the authors meant 0.6 to 1.0.

2.3 Data processing pre-checks

Beginning here the authors need greater clarity in their description of what was done.

Lines 256-260. What does background mean? It seems that an a priori assessment has to be made. Or are the light emitting city pixels included in the mean cloud-free background pixels?

Figure 2d. The green points are very hard to see. Would light blue be a better choice? Would it be better to use yellow for these pixels and green for signal saturation? Line 285 should be "retrievals"

Line 286, "attached" again. "Supplement"?

Figure 3. If the edges of the image correspond to the edges of the bounding box, please state so. If not, then draw the bounding box on the image.

Lines 295-304. In the discussion of cloud contamination and partly cloudiness there is no discussion of thin cirrus. There should be.

Line 309. Why the lopsided statistical cleaning? Dark Target aerosol algorithms do a symmetrical filtering over ocean in order to not impose a bias on the statistics. Over land the filtering is skewed because of bright surface pixels skew unfiltered biases towards too high AOD. Why the skewed filtering here?

Lines 321 – 325. I'm not clear on this. Are you finding the mean for each green pixel? For each city of many pixels, so that first there is a spatial mean of all the green pixels and then a temporal mean to get the average over the time span? Or is this a spatial and then temporal mean for all the green pixels within the bounding box? Is there a difference between green pixels within a bounding box and the green pixels within a "city". I have been assuming throughout the paper that standard deviation is a spatial standard deviation of all the green pixels either within a city or within a bounding box.

[Figure]

This is supported at line 386. But here it could be a temporal stdev or an ensemble stdev from all cities on the particular Julian Day.

Line 329. Should that be $\pm 0.1$X N_STD? plus or minus?

Line 346-349. This could be worded better. "non-totally cloudy" is clumsy. "non-overcast" is better. Though, "minimum number of non-overcast observations" is also clumsy. How about "June and July had the fewest acceptable observations".

Figures 4, 6,7,8,11 are not informative. How about density scatter plots? Or bin and average the data with standard deviation bars? Or calculate the regression statistics of those red lines in Figure 4? There will be additional comments on the scatter plot figures.

Lines 365-367. There is a lot of discussion in the paper and speculation about snow contamination with no evidence to support the speculation. Could snowy and non-snowy places (ie. California) be compared?

Lines 374 – 376. More on figures. That red line doesn't look like a good fit in Figures 4g and 4h, which makes the statements in these lines hard to accept. The red curves have a minimum around 30 deg, not nadir, so slant path length doesn't seem to be the best answer. This leaves anisotropic light sources. This is very interesting. Why?

Figure 5b. It would certainly be helpful to see some regression statistics.

Lines 399-408. The metric for DEL-Ia is complicated. The average stdev of the most variable 30% of the spatial stdevs, PLUS 2 stdevs of the stdevs of the most variable 30% of the spatial stdevs. I just don't see how this eliminates cloud contamination and lightning strikes. You have the top 1% of spatially variable light sources, but that top 1% will include clouds and lightning strikes. Also why can't you just use Rstd(1%) instead of finding mean and stdev of the top 30%.

Lines 460-461. Regression slope is mentioned here. I know that there is a line of thought in the community that wants to get away from regression statistics, and so

previously, even though I wanted to see regression statistics in the plots, I've refrained from demanding them. Except here regression slope is mentioned. If it is mentioned, then please show slopes, intercepts and R on the plots. If the regression is not linear, then show RMSE and R.

Lines 471-472. These criteria will eliminate places with highly variable day-to-day changes in AOT.

Figure 7 is crying for a color density plot.

Line 586. Sure for the U.S., but in other places the winter season might be fine.
* * *

---

## Referee Comment (RC2) · Anonymous Referee #2 · 8 Mar 2019

General comments:

The paper " Characterization and application of artificial light sources for nighttime aerosol optical depth retrievals using the VIIRS Day/Night Band" deals with opportunities to detect and characterize aerosol at night by measuring visible radiance from city lights in VIIRS Day-Night band. The topic meets the aim and scopes of the journal. The general structure of the paper is good.

The figures and its captions are of good quality.

The motivation and background are well reasoned in the introduction. It lists many relevant related papers. The presented research introduces novel concepts to use

the relatively new DNB channel on VIIRS to use visible radiance during the night for aerosol characterization. This, if successful, could close the nighttime gap of aerosol observations by satellite-based retrievals.

Section 2.2 describes the theoretical approach. The aerosol optical depth retrieval is based on a comparison of spatial standard deviation under a cloud and aerosol-free conditions to the measured spatial standard deviation. For practical results, the authors describe filtering and data selection processes in a sufficient way.

The result section shows that this approach has some skill, but also still major limitations. The authors discuss this in the concluding chapter and give suggestions for follow-up research.

I recommend minor revisions for the points specified in the next section.

Specific comments

Line 172: I can't find the full list of the cities in an attachment.

Line 305: "On each night and for each light source, the averaged radiance, its standard deviation.": What is here a light source? Can't be one individual DNB pixel. Is it a city (e.g. all green pixels in Fig.3b for Iowa City?

Line 305: Define lunar fraction.

Fig.11 and Equation 5: How can the transmission correction factor k as defined in Eq.5 be bigger than 1, as shown in Fig.11?
* * *

---

## Author Comment (AC1) · 30 Apr 2019

**Reviewer 1:**

Comments: This is the third paper from this group investigating the possibilities of deriving aerosol optical thickness at night from the VIIRS DNB. Interestingly I also reviewed the first paper in this series from 2013. I did not review or read the second paper in the series (McHardy et al., 2015) until I accepted this review. The group is making progress, but progress is slow. I would have hoped that they would have offered more than an incremental improvement to their 2015 paper, which is what we have here, and for awhile I was trying to determine whether this increment was sufficiently novel to warrant publication.

In this installment the authors implement a retrieval developed in the 2015 paper that uses the blurring of the spatial variability of light emitting city pixels at night to derive aerosol optical thickness (AOT). This is different from the paper that I reviewed in 2013 that used the contrast between the lit cities and the surrounding dark countryside. The idea of linking AOT to reduced sharpness of the image is old, and I am thinking of Tanré and Legrand (1991). The goal of a night time AOT from an imager is fundamentally ambitious. Everything from cloud clearing to the instability of artificial lights to the impossibility of collocating retrievals with current AERONET observations is working against you. Yet, the pay off is extremely rich, and as an aerosol product user, I am very happy that this group has not given up hope. The manuscript as it stands now could use some scrubbing to make it easier to understand, and I will make some recommendations. In some places the authors have made things complicated when simplicity should suffice. In particular the derivation of the cloud/aerosol free spatial variability measure for each target is complicated. It would also be helpful to start to quantify AOT accuracy using this method. Previous papers made that attempt, but here there was nothing. Overall the technique and implementation are sound, the presentation is sufficiently clean and the results sufficiently novel for publication. Therefore, I recommend publication after revision.

Tanré, D., & Legrand, M. (1991). On the satellite retrieval of Saharan dust optical thickness over land: Two different approaches. Journal of Geophysical Research: Atmospheres, 96(D3), 5221-5227.

*Response: We thank the reviewer for his/her constructive comments and suggestions. Also, the focus of this paper is to study the variations of artificial light sources for various observational conditions as well as explore the potential of applying the method for regional retrievals. Note that the theoretical analysis of retrieval errors has been reported in McHardy et al. (2015), as mentioned by the reviewer. To quantify uncertainties, besides already attempted theoretical approaches, empirical approaches may be needed. This requires the comparison of VIIRS AOT with a benchmark AOD dataset. However, both daytime AERONET and nighttime CALIOP AOD data have their own issues, as mentioned in the paper and are not the best sources for estimating uncertainties empirically. Note that nighttime AOT data from AERONET have only recently become available, after the submission of this paper. Thus, we leave the validation efforts using nighttime AERONET data for a future paper. In this study, we included discussions on limitations of the study, hopefully adding a different angle for uncertainties of the study.*

Point by point comments and recommendations.

Comments:  Introduction. There are a few other groups besides this one that have attempted similar aerosol retrievals using the DNB, or investigated the possibility theoretically. In particular
I recommend referencing Wang et al. (2016), and possibly Choo and Jeong (2016), though the latter is very local in its scope. Wang et al. (2016) put to rest the importance of variable water vapor in the method. If they hadn't, I would be worried about that factor.

Wang, J., Aegerter, C., Xu, X., & Szykman, J. J. (2016). Potential application of VIIRS Day/Night Band for monitoring nighttime surface PM2. 5 air quality from space. Atmospheric Environment, 124, 55-63.
Choo, G. H., & Jeong, M. J. (2016). Estimation of nighttime aerosol optical thickness from Suomi-NPP DNB observations over small cities in Korea. Korean Journal of Remote Sensing, 32(2), 73-86.

*Response: Thanks for the suggestions.  We have cited the two papers.*

Comments: Lines 88-95. I found the use of present tense "develops" (line 88) and "proposes" (line 90) a little odd. Those studies are done and published. I would have used "developed" and "proposed", but I'm not sure this is required.

*Response: Done.*

Comments: What is the time span or dates of the study? Is it one year of data or multiple years? I don't see this stated anywhere.
Likewise no where is it stated that this is specifically Suomi-NPP VIIRS, because there are now more than one VIIRS flying. There's NOAA 20, and over the years there will be more.

*Response: To avoid confusion, we have modified the following sentence to highlight the fact that only 2015 Suomi NPP VIIRS data were used.*
*"In this study, three processed and terrain-corrected Suomi NPP VIIRS datasets were used for 2015."*

Comments: The specific data sets look to me that they are IDPS data sets. IDPS processing and archiving of data sets has stopped or will stop very soon to be replace by EPS data sets and processing. I think this should be clarified in the text.

*Response: "VIIRS Cloud Cover Layers EDR (VCCLO) (public 04/27/2013)" and "VIIRS Day Night Band SDR (SVDNB) (public 02/07/2012)" were obtained between 9-2018 and 10-2018 from the Comprehensive Large Array-Data Stewardship System (CLASS) at https://www.avl.class.noaa.gov/saa/products.*

*We have added the following discussions in the paper.*
*The VIIRS data were obtained from the NOAA Comprehensive Large Array-Data Stewardship System (CLASS) site (https://www.avl.class.noaa.gov/saa/products)*

Comments: Line 172 "attachment". First should this be "attachment" or "supplements"? Note that
I was unable to down load or to see any of the supplementary files. This is likely a problem between me and the journal, not with the authors, but this should be tested.

*Response: We changed from "attachment" to "supplements".  Also, we checked MS Records for the paper and the above mentioned file was submitted to AMT on Dec. 14, 2018 as a supplement and is available for the corresponding author to download.  We are unsure why the reviewer can't see the file.   We will upload the file again..*

Comments: Line 182. "optical depth". Elsewhere the authors use the term "optical thickness". It would be good to be consistent.

*Response: Done.*

Comments: Line 219. I believe the authors meant 0.6 to 1.0.

*Response: Typo corrected.  Thanks.*

Comments: Beginning here the authors need greater clarity in their description of what was done. Lines 256-260. What does background mean? It seems that an a priori assessment has to be made. Or are the light emitting city pixels included in the mean cloud-free background pixels?

*Response: Background means nearby non-artificial light source pixels.   We have modified the sentence as:*
*"a given pixel to background pixels (non-artificial light pixels), as suggested in Johnson et al. (2013).  "*

Comments: Figure 2d. The green points are very hard to see. Would light blue be a better choice?  Would it be better to use yellow for these pixels and green for signal saturation?

*Response: Thank you for your suggestion. We made the changes. But since the cities are rather small we do not observe a significant improvement. Thus we kept the old figure.*

[Figure]

Comments: Line 285 should be "retrievals"

*Response: Done.*

Comments: Line 286, "attached" again. "Supplement"?

*Response: Done.*

Comments: If the edges of the image correspond to the edges of the bounding box, please state so. If not, then draw the bounding box on the image.

*Response: Thank you for your suggestion. We have added the revised figure, as suggested, to the paper.*

Comments: Lines 295-304. In the discussion of cloud contamination and partly cloudiness there is no discussion of thin cirrus. There should be.

*Response: We have revised the sentence to:*
*"Cloud contamination, especially cirrus cloud contamination, remains an issue in the above steps"*

Comments: Line 309. Why the lopsided statistical cleaning? Dark Target aerosol algorithms do a symmetrical filtering over ocean in order to not impose a bias on the statistics. Over land the filtering is skewed because of bright surface pixels skew unfiltered biases towards too high AOD. Why the skewed filtering here?

*Response: The lopsided statistical cleaning is applied as the histogram of radiance for a typical artificial light source is often non-Gaussian, and is biased towards lower radiance values. The non-Gaussian feature is the rationale behind a non-symmetrical filtering.*

Comments: Lines 321 – 325. I'm not clear on this. Are you finding the mean for each green pixel? For each city of many pixels, so that first there is a spatial mean of all the green pixels and then a temporal mean to get the average over the time span? Or is this a spatial and then temporal mean for all the green pixels within the bounding box? Is there a difference between green pixels within a bounding box and the green pixels within a "city". I have been assuming throughout the paper that standard deviation is a spatial standard deviation of all the green pixels either within a city or within a bounding box. This is supported at line 386. But here it could be a temporal stdev or an ensemble stdev from all cities on the particular Julian Day.

*Response: For each artificial light source, there are countable artificial light source pixels (green pixels). Thus, for each night, for each artificial light source, the mean and the standard deviation of those artificial light source pixels for the given artificial light source are computed. Then, yearly means are computed from those daily estimations. Also, we assume green pixels within a bounding box represent green pixels within a "city". We have added the following discussions to avoid confusion.*

*"Here, for each artificial light source (city/town), for a given satellite overpass of a given night, the mean radiance and the standard deviation of radiance for artificial light source pixels within the given city/town are computed, and are further used as the base elements for computing yearly mean radiance and standard deviation values."*

Comment: Line 329. Should that be _0.1X N_STD? plus or minus?

*Response: Here we used N-0.1X N_STD to screening out nights with very low artificial light source pixels, mostly likely due to cloud coverage.*

Comment: Line 346-349. This could be worded better. "non-totally cloudy" is clumsy. "nonovercast" is better. Though, "minimum number of non-overcast observations" is also clumsy. How about "June and July had the fewest acceptable observations".

*Response: We changed to "number of observations (cloud free or partially cloudy)"*

Comment: Figures 4, 6,7,8,11 are not informative. How about density scatter plots? Or bin and average the data with standard deviation bars? Or calculate the regression statistics of those red lines in Figure 4? There will be additional comments on the scatter plot figures.

*Response: We have added density scatter plots as suggested for Figures 4,6,7, and 8. For Figure 11, we reduced the symbol size. We have also added regression statistics in Figures 6,7, and 8.*

Comment: Lines 365-367. There is a lot of discussion in the paper and speculation about snow contamination with no evidence to support the speculation. Could snowy and nonsnowy places (ie. California) be compared?

*Response: Great question. We made the suggestion based on the figure below. Here we studied the normalized R-STD as a function of Julian day for three latitude rages (< 30°, 30-40°, and > 40°). The high spikes are mostly found over high latitudes during wintertime. However, to fully study the impact of snow covered surfaces on this study, the topic may deserve a study of its own and thus we leave this for a future paper.*

[Figure]

Comment: Lines 374 – 376. More on figures. That red line doesn't look like a good fit in Figures 4g and 4h, which makes the statements in these lines hard to accept. The red curves have a minimum around 30 deg, not nadir, so slant path length doesn't seem to be the best answer. This leaves anisotropic light sources. This is very interesting. Why?

*Response: The red lines in Figures 4g and 4h are computed through second order regression / polynomial curve fitting. The above mentioned patterns are likely introduced by function fitting through a dataset with noisy data points.*

Comment: Figure 5b. It would certainly be helpful to see some regression statistics.

*Response: Done. We added the regression relationship to the figure.*

Comment: Lines 399-408. The metric for DEL-Ia is complicated. The average stdev of the most variable 30% of the spatial stdevs, PLUS 2 stdevs of the stdevs of the most variable 30% of the spatial stdevs. I just don't see how this eliminates cloud contamination and lightning strikes. You have the top 1% of spatially variable light sources, but that top 1% will include clouds and lightning strikes. Also why can't you just use Rstd(1%) instead of finding mean and stdev of the top 30%.

*Response: We tried to answer this question based on our understanding of the question. As the reviewer suggested, the top 1% may include cloud and lightning contamination, and thus Rstd(1%) could be biased. The use of the most variable 30% of the spatial stdevs is attempted in an effort to reduce this issue. As with more data samples included, uncertainties due to cloud and lightning contamination may be averaged out, resulting in a more statistically stable DEL-Ia estimation.*

Comment: Lines 460-461. Regression slope is mentioned here. I know that there is a line of thought in the community that wants to get away from regression statistics, and so previously, even though I wanted to see regression statistics in the plots, I've refrained from demanding them. Except here regression slope is mentioned. If it is mentioned, then please show slopes, intercepts and R on the plots. If the regression is not linear, then show RMSE and R.

*Response: We have added regression statistics in the plots as suggested.*

Comment: Lines 471-472. These criteria will eliminate places with highly variable day-to-day changes in AOT.

*Response: We have added the following sentence to highlight the suggestion:*
*"Note that these criteria may exclude artificial light sources with highly variable day-to-day changes in AOT."*

Comment: Figure 7 is crying for a color density plot.

*Response:  Done.*

Comment: Line 586. Sure for the U.S., but in other places the winter season might be fine.

*Response: We agreed.  We have revised the sentence to:*
*"indicating the need for careful snow and ice detection for nighttime retrievals using VIIRS data for regions that may experience snow/ice coverage. "*

---

## Author Comment (AC3) · 30 Apr 2019

**Reviewer 2:**

Comments: The paper " Characterization and application of artificial light sources for nighttime aerosol optical depth retrievals using the VIIRS Day/Night Band" deals with opportunities to detect and characterize aerosol at night by measuring visible radiance from city lights in VIIRS Day-Night band. The topic meets the aim and scopes of the journal.
The general structure of the paper is good.
The figures and its captions are of good quality.
The motivation and background are well reasoned in the introduction. It lists many relevant related papers. The presented research introduces novel concepts to use the relatively new DNB channel on VIIRS to use visible radiance during the night for aerosol characterization. This, if successful, could close the nighttime gap of aerosol observations by satellite-based retrievals.
Section 2.2 describes the theoretical approach. The aerosol optical depth retrieval is based on a comparison of spatial standard deviation under a cloud and aerosol-free conditions to the measured spatial standard deviation. For practical results, the authors describe filtering and data selection processes in a sufficient way.
The result section shows that this approach has some skill, but also still major limitations.
The authors discuss this in the concluding chapter and give suggestions for follow-up research.
I recommend minor revisions for the points specified in the next section.

*Response: We thank the reviewer for his/her comments and suggestions.*

Specific comments

Comments:  Line 172: I can't find the full list of the cities in an attachment.

*Response: We checked MS Records for the paper and the above mentioned file was submitted to AMT on Dec. 14, 2018 (supplement) and is available for the corresponding author to download. We are unsure why the reviewer can't see the file.   We will upload the file again.*

Comments:  Line 305: "On each night and for each light source, the averaged radiance, its standard deviation.": What is here a light source? Can't be one individual DNB pixel. Is it a city (e.g. all green pixels in Fig.3b for Iowa City?

*Response: Yes, all green pixels in Fig. 3b represent valid artificial light source pixels for a given city. We have revised the sentence as:*
*"On each night and for each light source (e.g., a given city that is composed of multiple VIIRS DNB pixels such as shown in Fig. 3b), the averaged radiance, its standard deviation, the lunar fraction (fraction of the lunar disk illuminated by the sun, as viewed from Earth), viewing geometries, and the number of artificial light source pixels identified, are reported as diagnostic information."*

Comments:   Line 305: Define lunar fraction.

*Response: We have revised the sentence as "the lunar fraction (fraction of the lunar disk illuminated by the sun, as viewed from Earth)"*

Comments:   Fig.11 and Equation 5: How can the transmission correction factor k as defined in Eq.5 be bigger than 1, as shown in Fig.11?

*Response: It is introduced by uncertainties in various factors, such as uncertainties in $\Delta I_a$ and CALIOP AOT.*